

# Serum β-klotho is a potential biomarker for diagnosing alcoholic liver disease and differentiating from nonalcoholic fatty liver disease

Chengmei Fang[1,2,*], Xin Miao[3,*], Chuyan Peng[2,4], Zhenguo Xie[5], Fuzhen Zhao[4], Tian Chen[6], Wenjin Zhang[2], Xiaofei Bi[7], Xuan An[2,4,7] and Guicheng Wu[1,2,4,7]

[1] Department of Infectious Disease, Southwest Medical University, Sichuan, China

[2] Chongqing Municipality Clinical Research Center for Endocrinology and Metabolic Diseases, Chongqing University Three Gorges Hospital, Chongqing, China

[3] Blood Transfusion Department, Chongqing University Three Gorges Hospital, Chongqing, China

[4] School of Medicine, Chongqing University, Chongqing, China

[5] Department of Pharmacy, Chongqing University Three Gorges Hospital, Chongqing, China

[6] Health Management Center, Chongqing University Three Gorges Hospital, Chongqing, China

[7] Department of Hepatology, Chongqing University Three Gorges Hospital, Chongqing, China

[*] These authors contributed equally to this work.

Corresponding authors
Xuan An, 57485971@qq.com
Guicheng Wu,
wuguicheng@cqu.edu.cn

## ABSTRACT

**Background.** Alcoholic liver disease (ALD), with the control of infectious liver disease and the improvement in living standards, is emerging as a significant liver disease posing a threat to public health. Besides, ALD often overlaps or coexists with nonalcoholic fatty liver disease (NAFLD), however, due to the lack of specific non-invasive biomarkers and the fact that drinkers' self-reported alcohol consumption is often concealed, the identification of ALD and NAFLD is sometimes not easy. This study aims to explore a new specific serum biomarker to more easily diagnose ALD and differentiate it from NAFLD.

**Subjects and Methods.** A total of 204 serum samples were collected, including 70 from ALD patients, 68 from NAFLD patients and 66 from healthy controls (HC). Serum β-klotho (sKLB) levels were measured using the enzyme-linked immunosorbent assay (ELISA). The diagnostic performance of potential biomarkers was evaluated using the area under the receive operating characteristic curve (AUROC).

**Results.** The levels of sKLB were significantly elevated (1,332.12 (410.40, 2,687.00) pg/mL, $p < 0.001$) in ALD patients and significantly reduced in NAFLD patients (47.82 (32.76, 77.11) pg/mL, $p = 0.018$) compared to the healthy controls. The AUROC for sKLB in diagnosing ALD is 0.927, which was higher than that for the aspartate aminotransferase (AST)/alanine aminotransferase (ALT) ratio (0.672) and γ-glutamyl transpeptidase (GGT) (0.891). The combined AUROC for sKLB + AST/ALT, sKLB + GGT, and AST/ALT ratio + GGT in diagnosing ALD were 0.924, 0.967 and 0.917, respectively.

**Conclusion.** sKLB is a potential biomarker for diagnosing ALD, and may aid in differentiating between ALD and NAFLD, when combined with GGT, sKLB offers enhanced diagnostic sensitivity and specificity for ALD.

## INTRODUCTION

With the effective prevention and control of infectious liver diseases such as hepatitis B and hepatitis C (*Te & Jensen, 2010*; *Liu et al., 2019*), non-infectious liver diseases, including alcoholic liver disease (ALD) and non-alcoholic fatty liver disease (NAFLD) have become the predominant contributors to chronic liver disease globally (*Younossi et al., 2023*). ALD is emerging as a significant public health problem due to its substantial disease burden and extensive socioeconomic impact (*Cao & Fan, 2019*; *Díaz et al., 2023*). Given that alcohol abuse , obesity and metabolic syndrome often co-exist in some patients the number of ALD co-existing with NAFLD is increasing (*Rowell & Anstee, 2015*; *Cao & Fan, 2019*; *Díaz et al., 2023*). ALD is caused by alcohol toxicity, while NAFLD is caused by metabolism abnormalities (*Idalsoaga et al., 2020*). Early identification relies on a detailed drinking history and metabolic assessment, but most patients have vague descriptions of their alcohol consumption, AST/ALT $\geq$ 2 (*Kalas et al., 2021*) or increased gamma-glutamyl transpeptidase (GGT) are often used to diagnose ALD, but the specificity is poor (*Gowda et al., 2009*; *Gómez-Medina et al., 2023*), which poses certain challenges to clinical diagnosis. Therefore, it is particularly important to find a new biomarker to assist in the differentiation of ALD and NAFLD.

β-klotho (KLB) is a 1,043-amino-acid single-pass transmembrane protein widely expressed in the liver, adipose tissue, pancreas and gut (*Kuro-o, 2012*). As a co-receptor of fibroblast growth factor (FGF) 21/19, KLB plays critical roles in energy metabolism and cell signaling (*Aaldijk et al., 2023*). Genome-wide association studies (GWAS) have identified KLB single nucleotide polymorphisms (SNPs) associated with alcohol consumption in human (*Sanchez-Roige et al., 2019*). Among these, the rs17618244 variant has been linked to hepatic ballooning, fibrosis, inflammation, and cirrhosis in pediatric and obese NAFLD patients (*Panera et al., 2021*). Recent studies have revealed the key role of KLB in regulating lipid metabolism and bile acid (BA) metabolism (*Somm et al., 2018*), KLB knockout mice exhibit altered BA composition, hepatic inflammation, and early fibrosis, indicating that KLB deficiency disrupts the liver-gut BA circuit and induces lipid metabolism disorders (*Somm et al., 2018*). In ketogenic diet models, KLB deficiency impairs FGF21 signaling, negating the beneficial effects of the ketogenic diets on fatty liver by inhibiting fatty acid oxidation and enhancing lipogenesis (*Guo et al., 2024*). In ALD mouse models, upregulating intestinal KLB expression mitigates ethanol-induced liver damage and inflammation, suggesting that a protective role *via* modulation of the gut-liver axis (*Hou et al., 2022*).

*In vivo*, KLB exists in two forms: secreted KLB (sKLB, also known as soluble KLB or serum KLB), which comprises only the extracellular protion and arises from gene mutations or proteolytic cleavage (*Lee et al., 2018a*), and membrane-bound KLB (mKLB), which includes intracellular and extracellular domains (*Kuzina et al., 2019*). While the role of sKLB in ALD has not been previously explored, we hypothesize that chronic alcohol consumption may affect KLB expression or shedding, thereby altering sKLB levels. sKLB

may therefore represent a novel and disease-specific biomarker for ALD. This study aims to investigate the role of sKLB in ALD and asses its potential utility in distinguishing between ALD and NAFLD.

## MATERIALS AND METHODS

### Study population

The sample size was calculated based on the sensitivity and specificity derived from preliminary experimental results. The formula used was $N = Z^2_{1-\alpha/2}{}^{\star}p(1-p)/\delta2$, where N represents the sample size, $\delta$ is the allowable error, and p denotes sensitivity or specificity. Specificity was estimated at 90% ± 10%, and sensitivity at 80% ± 10%, with an allowable error was 0.1. The significance level ($\alpha$) was set at 0.05 for a two-tailed test. Based on these parameters, the required sample sizes for specificity and sensitivity were calculated as 62 and 35, respectively. Consequently, a minimum of 62 participants was determined to be necessary. In total 70 patiens with ALD, 68 with NAFLD and 66 healthy controls (HC) were consecutively enrolled between May 2019 and August 2022 from the Department of Hepatology, and Health Management Center of Chongqing University Three Gorges Hospital. The study was approved by the Medical Ethics Committee of Chongqing University Three Gorges Hospital (approval number: 2022KY13). This trial is registered at medicalresearch.org, under registration number MR-50-24-013402. All participants provided written informed consent (see Supplemental Information).

Healthy controls (HCs) were individuals who abstained alcohol throughout life or consumed alcohol occasionally, with intake levels below the diagnostic threshold for ALD and no history of regular or long-term heavy drinking. They were of major health issues, defined as having normal complete blood count and biochemistry, negative for viral hepatitis markers, normal chest X-ray and abdominal ultrasound, and no history of chronic diseases such as hypertension, type 2 diabetes mellitus, and without other chronic liver diseases such as autoimmune hepatitis, virus hepatitis, alcohol-related liver disease, nonalcoholic fatty liver disease, or any diagnosed malignancy.

Alcohol consumption in patients with ALD was assessed by qualified physician based on clinical evidence and medical history. Inclusion criteria required a daily alcohol intake of more than 20 g for females and 40 g for males, with a drinking history of more than 5 years. In addition, patients had to exhibit clinical and/or biological evidence of live injury, in accordance with the guidelines of European Association for the Study of the liver (*European Association for the Study of the Liver, 2018*), the alcohol consumption of each patient was assessed by the Alcohol Use Disorders Identification Test-Consumption (AUDIT-C) questionnaire. Patients with metabolic disorders, malignancies, coronary artery disease, viral hepatitis, drug-induced liver injury, or autoimmune liver disease were excluded from the study.

The diagnosis of NAFLD was based characteristic findings from abdominal color ultrasonography, including enhanced front-field echo, attenuated far-field echo, and indistinct intrahepatic duct structures,as per the guideline of the European Association for the Liver (EASL), the European Association for the Study of Diabetes (EASD), and the

European Association for the Study of Obesity (EASO) (*European Association for the Study of the Liver (EASL), European Association for the Study of Diabetes (EASD) & European Association for the Study of Obesity (EASO), 2016*). Patients with a history of excessive alcohol consumption or other identifiable causes of fatty liver disease were excluded.

## Detection of sKLB and other biochemical indicators

Venous blood samples were collected from subjects after overnight fasting, between 8:00 and 12:00 in the morning. The freshly drawn blood was placed into vacuum blood collection tubes without additives and allowed to stand at room temperature for 20 min. Samples were then centrifuged at 4 °C and 3,000 rpm for 20 min. The resulting serum supernatant was transferred into sterile 1.5 mL EP tubes and immediately stored at −80 °C until all samples were collected for bath analysis. sKLB levels were measured using an ELISA kit (Catalog Number: DY5889-05; R&D Systems, Minneapolis, MN, USA). Prior to assay, serum samples from both HC and patients were thawed at room temperature for at least 15 min. A capture antibody was applied to a 96-well plate and incubated at room temperature for 12 h overnight. After washing the plate three times with washing buffer, 100uL of serum sample was added to each well and incubated for 2 h to allow binding. Following a washing step, HRP-conjugated detection antibody and luminescent substrate were added. Abosorbance was measured at 450nm using a Spectra Max M4 microreader (serial number: 21300111), with correction wavelength set at 570 nm, A calibration curve was calculated using ELISA cal software. Other biochemical indicators were measured using automated biochemical analyzers.

## Statistical analysis

Statistical analyses were performed using SPSS 25 (IBM Corp., Armonk, NY, USA), and graphical visualizations were created with GraphPad Prism version 10.0. Continuous variables flowing a normal distribution were presented as mean ± standard deviation (SD), and differences between two or more groups were evaluated by $t$-test or analysis of variance (ANOVA). Non-normal distribution were presented as median (Q1, Q3), and the intergroups comparisons were performed using Kruskal–Wallis test, followed by Bonferroni correction for *post hoc* analyses. ROC curve analysis and propensity score matching were performed using R4.4.0, with a matching tolerance of 0.1. Age, sex, and body mass index (BMI) were included as covariates, and the propensity scores were calculated at a 2:1 ratio. To evaluate the diagnostic performance of the identified potential biomarkers, the receiver operating characteristic (ROC) curves were generated using MedCal, and the area under the curve (AUC) and its 95% confidence interval were estimated using the DeLong method, combined diagnostic performance was assesed using a binomial logistic regression model. Cutoff values, sensitivity, and specificity were determined based on the Youden index (sensitivity+specificity−1). To evaluate model robustness, bootstrap validation with 2,000 resampling iterations was employed during the training phase. A $p$-value $< 0.05$ was considered statistically significant.

**Table 1 Clinical characteristics of the HC, the NAFLD and the ALD groups.**

|  | HC = 66 | NAFLD = 68 | ALD = 70 | P |
|---|---|---|---|---|
| Sex = Female, n(%) | 19 (28.8 ) | 10 (14.7 ) | 6 (8.6 ) | 0.006 |
| Age | 52 (49, 54) | 53 (51, 55) | 58 (51, 68) | <0.001 |
| GLU | 5.60 (1.35) | 5.92 (0.99) | 6.42 (2.20) | 0.013 |
| TG | 1.28 (0.54) | 2.11 (0.98) | 1.49 (0.92) | <0.001 |
| CHOL | 5.04 (0.85) | 5.06 (0.98) | 4.19 (1.46) | <0.001 |
| LDL | 3.21 (0.76) | 3.31 (0.89) | 2.43 (1.13) | <0.001 |
| HDL | 1.46 (0.31) | 1.17 (0.26) | 1.13 (0.51) | <0.001 |
| CREA | 78.82 (14.65) | 83.49 (17.28) | 88.67 (61.17) | 0.323 |
| ALB | 47.53 (2.09) | 47.54 (3.77) | 35.69 (6.40) | <0.001 |
| TBIL | 13.43 (5.48) | 11.95 (5.28) | 69.71 (104.94) | <0.001 |
| GGT | 29.03 (22.48) | 51.69 (47.42) | 283.14 (349.51) | <0.001 |
| ALT | 18.68 (9.89) | 26.14 (11.70) | 135.03 (360.62) | 0.002 |
| AST | 19.13 (4.83) | 22.84 (7.39) | 221.61 (1099.99) | 0.11 |
| ALP | 74.89 (20.06) | 80.07 (17.96) | 163.47 (133.49) | <0.001 |
| PLT | 208.82 (60.40) | 218.03 (58.60) | 128.23 (87.32) | <0.001 |

**Notes.**

Continuous variables are presented as median (IQR), depending on their distribution, while categorical variables are expressed as counts (n, %).

Abbreviations: GLU, glucose; TG, triglyceride; CHOL, cholesterol; LDL, low-density lipoprotein; HDL, high-density lipoprotein; CREA, creatinine; ALB, albumin; TBIL, total bilirubin; GGT, γ-glutamyl transpeptidase; ALT, alanine aminotransferase; AST, aspartate aminotransferase; ALP, alkaline phosphatase; PLT, platelet count; IQR, interquartile range; HC, healthy control; NAFLD, non-alcoholic fatty liver disease; ALD, alcoholic liver disease.

# RESULTS

## General demographic characteristics

Clinical data were obtained from the medical records. In the ALD group, participants ranged in age from 51 to 68 years; in the NAFLD group, from 51 to 55 years; and in the HC group, from 49 to 55 years. The majority of subjects in the three groups were male. The ALD group encompassed the full spectrum of ALD, including steatosis, inflammation, and fibrosis. Demographic and baseline clinical characteristics for the three groups were summarized in Table 1. Compared to the HC and NAFLD groups, the ALD group exhibited significant differences in most serum biochemical indices as well as in age and sex. To account for these differences in subsequent analyses, propensity matching was performed based on age, gender, and BMI. The characteristics after propensity matching were provided in the supplementary data Tables S1 and S2.

## sKLB levels were increased in ALD and decreased in NAFLD, unlike AST/ALT ratio and GGT levels in ALD

The ALD group echibited the highest sKLB levels (1,332.12 (410.40, 2,687.00) pg/mL), while the NAFLD group showed the lowest sKLB levels (47.82 (32.76, 77.11) pg/mL, Fig. 1A). Compared to the HC group (74.63 (50.53, 180.20) pg/mL), sKLB levels were significantly higher in the ALD group ($p < 0.0001$). Conversely, sKLB levels in the NAFLD group were significantly lower than those in the ALD group ($p < 0.0001$) and HC group ($p = 0.018$) group. The AST/ALT ratio was significantly increased in the ALD group

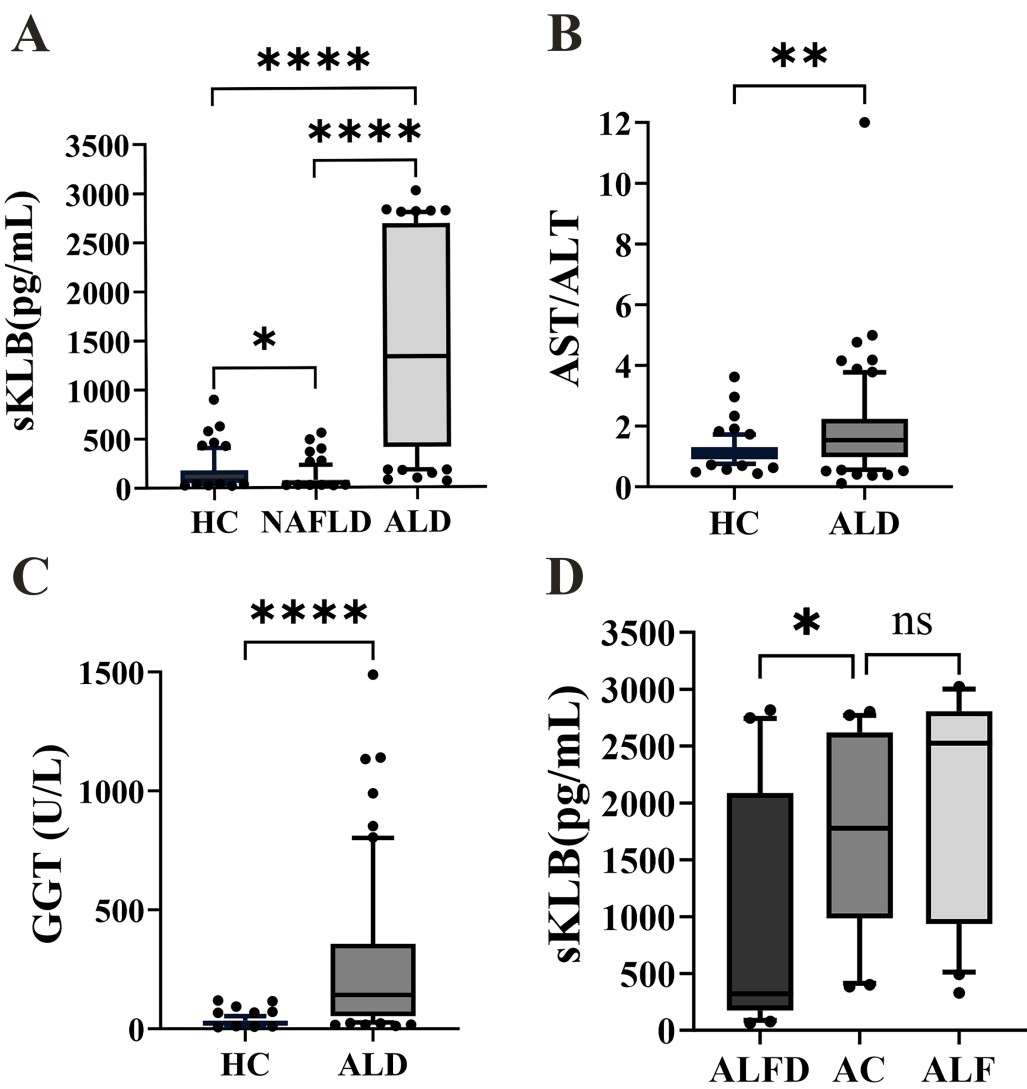

**Figure 1** **The sKLB, GGT level and AST/ALT ratio expression in the difference groups.** (A) sKLB level expression in HC, NAFLD, and ALD groups, $N = 66, 68, 70$; (B) AST/ALT ratio expression in HC and ALD groups, $N = 66, 70$; (C) GGT level expression in HC and ALD groups, $N = 66, 70$; (D) sKLB level expression in AFLD, AC and ALF groups, $N = 25, 25, 20$. ALFD, alcoholic fatty liver disease; AC, alcoholic cirrhosis; ALF, alcoholic liver failure. Data are depicted as box-and-whisker plots, illustrating the distribution of values within each group. The central horizontal line within each box signifies the median. The upper and lower boundaries of the box denote the 75th and 25th percentiles (quartiles), respectively; the whiskers extend to the 10th and 90th percentiles and the dots represent the outliers. The median represents the middle value of the data, while the interquartile range represents the middle 50% of the data. *$p < 0.05$; **$p < 0.01$; ****$p < 0.0001$; ns, not significant.

**Table 2 The relationship between sKLB and other clinical indicators in ALD.**

| Feature | sKLB | |
|---|---|---|
| | r | p |
| TBIL | 0.269 | 0.024 |
| TBA | 0.275 | 0.021 |
| CG | 0.181 | 0.134 |
| PLT | −0.169 | 0.161 |
| INR | 0.317 | 0.008 |
| HA | 0.251 | 0.036 |
| CIV | 0.246 | 0.040 |
| LN | 0.233 | 0.053 |
| HDL | −0.234 | 0.051 |
| AST | 0.236 | 0.049 |
| ALT | 0.083 | 0.492 |

Notes.

Correlation between sKLB expression and other serological indicators in ALD patients, using Spearman correlation coefficient evluated association strength.

Abbreviations: TBIL, total bilirubin; TBA, total bile acid; CG, cholyglycine; PLT, platelet; INR, international normalized ratio; HA, serum hyaluronic acid; CIV, type IV collagen; LN, laminin; HDL, high-density lipoprotein; AST, aspartate aminotransferase; ALT, alanine amiotransferase.

(1.533 (0.98,2.24)) compared to the HC group (1.103 (0.91,1.31), $p = 0.0012$) (Fig. 1B). Similarly, GGT levels were markedly higher in the ALD group (142.50 (54.00, 357.50) U/L, $p < 0.0001$) than in the HC group (22.50 (15.00, 34.25) U/L) (Fig. 1C). Among ALD subtypes, sKLB levels increased progressively: alcoholic fatty liver disease (AFLD) group (322.3 (177.4, 2,087) pg/mL), alcoholic cirrhosis (AC) group (1,777 (985.4, 2,619) pg/mL), and alcoholic liver failure (ALF) group (2,526 (934.2, 2,807) pg/mL) (Fig. 1D).

## Relationship between sKLB levels and other serological indicators

sKLB levels were positively correlated with TBIL, TBA, HA, CIV, AST and prothrombin time international normalized ratio (PT-INR, $p < 0.05$). A negatively correlation was observed with HDL, although this was marginally significant ($p = 0.051$) (Table 2).

## Diagnostic value of sKLB in ALD

The AUROCs for sKLB in diagnosing ALD was 0.927, with a sensitivity of 80%, specificity of 87.9% at a cut-off value of 379.5 pg/mL. This performance surpassed that of the AST/ALT ratio (AUROC = 0.672, sensitivity: 61.4%, specificity: 78.8%, cut-off = 1.3) and GGT (AUROC = 0.891, sensitivity: 75.7%, specificity: 90.9%, cut-off = 52.5). Among these three indicators, only sKLB achieved both sensitivity and specificity above 80%, highlighting its superior diagnostic accuracy for ALD, (Fig. 2). These results were consistent with even after propensity score matching (Fig. S1).

## Diagnostic value of sKLB combination with GGT or AST/ALT ratio in ALD

To evaluate whether combining sKLB with other biomarkers could enhance diagnostic performance, we calculated AUROCs for various combinations. For distinguishing ALD from HC, the AUROC combinations were as follow: sKLB + AST/ALT ratio: 0.924; sKLB

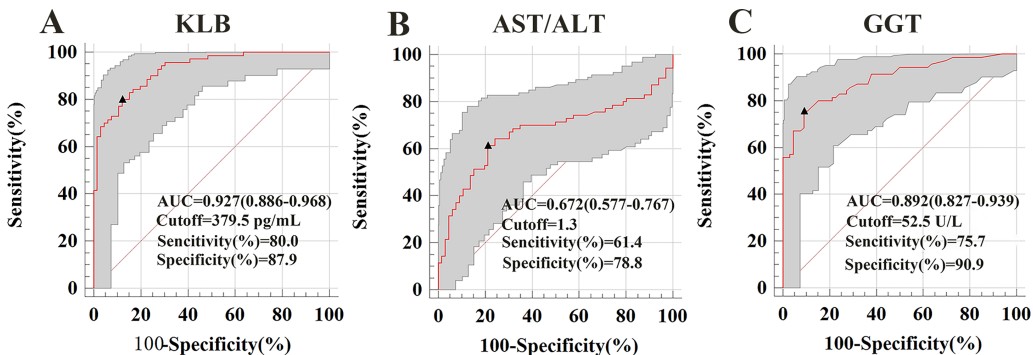

**Figure 2** Serum markers associated with ALD were used to construct a receiver operating characteristic (ROC) curve. The ROC curve demonstrates the diagnostic performance of serum markers (sKLB, AST/ALT, and GGT) in ALD. (A) ALD of sKLB: AUROC (the areas under the ROC curve) = 0.927 (95% CI [0.886–0.968]), sensitivity = 80.0%, specificity = 87.9%; (B) ALD of AST/ALT ratio: AUROC = 0.658 (95% CI [0.577–0.767]), sensitivity = 61.4%, specificity = 78.8%; (C) ALD of GGT: AUROC = 0.867 (95% CI [0.838–0.946]), sensitivity = 75.7%, specificity = 90.9%.

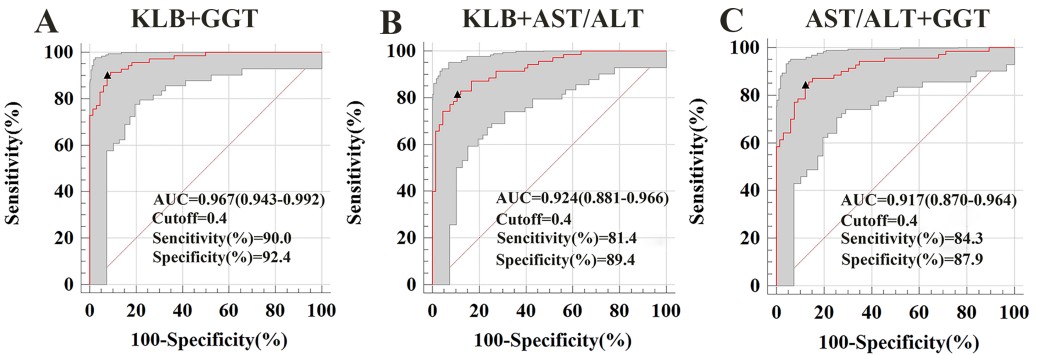

**Figure 3** Receiver operating characteristic curve for s KLB + AST/ALT, sKLB + GGT and AST/ALT + GGT in the ALD group. (A) ALD of sKLB + AST/ALT ratio: AUROC = 0. 924 (95% CI [0.881–0.966]), sensitivity = 81.4%, specificity = 89.4%; (B) ALD of sKLB + GGT: AUROC = 0.967 (95% CI [0.943–0.992]), sensitivity = 90.0%, specificity = 92.4%; (C) ALD of AST/ALT ratio + GGT: AUROC = 0.917 (95% CI [0.870–0.964]), sensitivity = 84.3%, specificity = 87.9%.

+ GGT:0.967; AST/ALT + GGT: 0.917. Among these, the combination of sKLB and GGT yielded the highest diagnostic accuracy (Fig. 3).

## DISCUSSION

KLB is a co-receptor of FGF21/FGF19 and is mainly involved in glucose and lipid metabolism and bile acid metabolism. Previous studies have found that FGF21 (*Huai et al., 2024*) and KLB play an important role in the treatment of alcoholic liver disease (*Hou et al., 2022*). In addition, alcohol-induced inflammation and fibrosis can enhance the expression of FGF21 (*Lee et al., 2018b*). It is speculated that sKLB, as a soluble form of FGF21 co-receptor, may reflect changes in the liver FGF signaling axis and the inflammatory

fibrotic state. It may be an intrinsic marker of the ALD pathological process and has unique advantages in reflecting the pathogenesis and early diagnosis of ALD.

In our study, sKLB levels were significantly increased in ALD patients compared with HC, demonstrating high diagnostic accuracy with an AUROC of 0.927 (sensitivity = 80%, specificity = 87.9%). Moreover, combining sKLB with GGT further enhanced diagnostic performance (AUROCs = 0.967, sensitivity = 90%, specificity = 92.4%). Although sKLB alone shows strong diagnostic accuracy for ALD, routine liver function markers such as AST/ALT ratio and GGT remain widely used. When combined with these conventional biomarkers, sKLB improved the diagnostic performance for ALD, with the combination with GGT yielding the highest sensitivity and specificity (Fig. S2 and Table S3). sKLB thus emerges as promising non-invasive biomarker for diagnosing ALD, It offers advantages over liver biopsy, including being low-risk and suitable for large-scale screening and longitudinal monitoring. Unlike AST/ALT ratio and GGT, sKLB levels differ significantly between ALD and NAFLD, enhancing its utility in differential diagnosis. Nonetheless, inter-individual and ethic variations may influence sKLB levels, underscoring the need for large, more diverse cohort studies.

Previous studies have identified the KLB rs17618244 G > A polymorphism as a genetic variant associated with reduced hepatic KLB expression and greater histological severity in NAFLD, including lobular inflammation, ballooning and fibrosis (*Panera et al., 2021*). *Lee et al. (2018b)* demonstrated that pro-inflammatory cytokines (IL-1β and TNF-α) suppress KLB expression through the NF-κB-JNK pathway in hepatocytes. In high-fat diet-induced NAFLD mouse models, inflammation and lipotoxicity promoted DNMT1/3A-mediated methylation of the KLB promoter, reducing KLB gene expression (*Wang et al., 2023*). Consistently, we observed significantly lower sKLB levels in the NAFLD group compared with HCs, likely reflecting a combination of transcriptional repression, epigenetic modification, and genetic predisposition. Conversely, the elevation of sKLB in ALD may reflect a stress response to hepatocellular injury, possibly mediated by ADAM17-dependent cleavage and shedding of the KLB ectodomain (*Kim et al., 2015*). ALD subgroup analysis showed that sKLB levels were significantly elevated in AC group compared with AFLD group. Although ALF group compared with AC group analysis did not reach statistical significance, sKLB levels tended to increase progressively from AFLD to AC and ALF groups, suggesting a possible correlation with ALD severity. This warrants further validation in larger cohorts.

The AST/ALT ratio is a well-established marker in ALD (*Kim & Park, 2020*), with AST levels typically elevated due to mitochondrial damage from ethanol metabolism (*Alatalo et al., 2009*; *Kwo, Cohen & Lim, 2017*; *Aulbach & Amuzie, 2017*). Additionally, chronic alcohol consumption can cause phosphate pyridoxine deficiency, disproportionately reducing ALT relative to AST, thereby increasing the AST/ALT ratio (*Robles-Diaz et al., 2015*). These findings align with our results. We also observed a positive correlation between sKLB and AST, suggesting that sKLB may be linked to mitochondrial stress response.

GGT, a transmembrane enzyme involved in amino acid transport, is primarily hepatic origin (*Corti et al., 2020*). Alcohol-induced hepatocellular damage causes GGT to leak into the circulation, increasing serum levels (*Van Beek et al., 2014*; *Hernandez-Tejero,*

*Clemente-Sanchez & Bataller, 2023*). However, we found no significant correlation between sKLB and GGT in ALD. In contrast, in our previous study of hepatitis B virus-related liver disease, sKLB was significantly correlated with GGT (*Miao et al., 2024*). Notably, sKLB levels in hepatitis B virus-related liver diseases correlated with disease severity but not with viral load, suggestion that the interaction between sKLB and GGT may differ depending on disease etiology (*Miao et al., 2024*). In ALD, appears to function independently of GGT, supporting its role as a distinct diagnostic biomarker.

Additionally, sKLB showed significant positive correlations with markers of cholestasis and fibrosis, including TBIL, TBA, HA, CIV. Chronic cholestatic conditions are known to promote liver fibrosis and cirrhosis if untreated (*Petrescu & DeMorrow, 2021*). Thus, elevated sKLB levels in ALD may reflect fibrotic remodeling, reinforcing its potential as a non-invasive biomarker for disease severity assessment.

Given that the liver is the primary site of HDL synthesis (*Han et al., 2021*), decreased HDL may indicate hepatic dysfunction (*Rao et al., 2021*). Our study observed a significant negative correlation between sKLB and HDL in ALD patients, suggesting a potential role for sKLB in dysregulated lipid metabolism and impaired reverse cholesterol transport during alcohol-induced liver injury.

Chronic ethanol exposure induces endoplasmic reticulum (ER) and oxidative stress in hepatocytes, leading to a robust upregulation of hepatic FGF21 as a protective response (*Wang, Farokhnia & Leggio, 2022*). This stress response, mediated *via* ATF4, may also transiently elevate KLB expression (*Dong et al., 2015*). Moreover, alcohol-induced inflammatory activates metalloproteases such as ADAM17, which cleaves membrane-bound KLB to release sKLB (*Kim et al., 2015*). Additional sKLB may be derived from vesicular shedding or alternative splicing due to cellular injury. Examining ADAM17 inhibition and KLB trafficking could help elucidate these mechanisms. As a co-receptor for FGF21/19 (*Schumann et al., 2016*), elevated sKLB may represent a compensatory or self-protective response during alcohol-induced hepatic injury.

This study's major strength lies in its clinical relevance—exploring sKLB as a diagnostic marker for ALD and its ability to distinguish ALD from NAFLD. However, some limitations must be acknowledged. First, the cohort consisted exclusively of Asian individuals. Given known inter-ethnic variations in alcohol-metabolizing enzymes such as ALDH2 and ADH1B (*Edenberg & McClintick, 2018*; *Wang et al., 2021*), extrapolation to other populations should be done cautiously. Cultural differences in drinking patterns— moderate drinking with meals in China *vs.* binge drinking in Western countries (*Lu et al., 2004*; *Liangpunsakul, Haber & McCaughan, 2016*)—may also influence disease development and biomarker expression. Broader studies involving diverse populations and drinking behaviors are necessary. Second, the majority of enrolled subjects were male due to the lower prevalence of ALD in Chinese females; future studies should aim for gender balance. Lastly, sKLB levels were assessed at a single time point. Longitudinal monitoring may better capture dynamic changes during disease progression or resolution.

## CONCLUSION

In conclusion, this study demonstrated that sKLB is a promising non-invasive biomarker for diagnosing ALD and differentiating it from NAFLD. When combined with traditional markers such as the AST/ALT ratio or GGT—particularly GGT—diagnostic accuracy is further enhanced. These findings support the potential clinical utility of sKLB for early detection, monitoring, and possibly stratifying disease severity in ALD.

## ACKNOWLEDGEMENTS

The authors would like to express gratitude to all of the staff who assisted in our research.

### Funding

This work was supported by the Chongqing Municipality Clinical Research Center for Endocrinology and Metabolic Diseases Construction Fund (No. 66002), the National Natural Science Foundation of China (Grant No 81873571), Chongqing Natural Science Foundation (Grant No cstc2019jcyjmsxmX0774), Beijing Science and Technology Innovation Medical Development Foundation (Grant No KC2021-JX-0186-128), the hospital-level scientific research project of Chongqing University Three Gorges Hospital (Grant No 2022YJKYXM-014) and the application study of standardized diagnosis and treatment process of abnormal liver function (2023TIAD-ZXX0015). The funders had no role in study design, data collection and analysis, decision to publish, or preparation of the manuscript.

### Grant Disclosures

The following grant information was disclosed by the authors:
Chongqing Municipality Clinical Research Center for Endocrinology and Metabolic Diseases Construction Fund: 66002.
National Natural Science Foundation of China: 81873571.
Chongqing Natural Science Foundation: cstc2019jcyjmsxmX0774.
Beijing Science and Technology Innovation Medical Development Foundation: KC2021-JX-0186-128.
Hospital-level scientific research project of Chongqing University Three Gorges Hospital: 2022YJKYXM-014.
Application study of standardized diagnosis and treatment process of abnormal liver function: 2023TIAD-ZXX0015.

### Competing Interests

The authors declare there are no competing interests.

### Author Contributions

- Chengmei Fang conceived and designed the experiments, performed the experiments, prepared figures and/or tables, authored or reviewed drafts of the article, and approved the final draft.

- Xin Miao conceived and designed the experiments, performed the experiments, authored or reviewed drafts of the article, and approved the final draft.
- Chuyan Peng performed the experiments, analyzed the data, authored or reviewed drafts of the article, and approved the final draft.
- Zhenguo Xie analyzed the data, prepared figures and/or tables, and approved the final draft.
- Fuzhen Zhao analyzed the data, authored or reviewed drafts of the article, and approved the final draft.
- Tian Chen conceived and designed the experiments, analyzed the data, prepared figures and/or tables, authored or reviewed drafts of the article, and approved the final draft.
- Wenjin Zhang analyzed the data, prepared figures and/or tables, and approved the final draft.
- Xiaofei Bi analyzed the data, prepared figures and/or tables, and approved the final draft.
- Xuan An conceived and designed the experiments, authored or reviewed drafts of the article, and approved the final draft.
- Guicheng Wu conceived and designed the experiments, authored or reviewed drafts of the article, and approved the final draft.

## Human Ethics

The following information was supplied relating to ethical approvals (i.e., approving body and any reference numbers):

The studies involving human participants were reviewed and approved by the medical ethics committee of Chongqing University Three Gorges Hospital (2022KY13).

## Clinical Trial Ethics

The following information was supplied relating to ethical approvals (i.e., approving body and any reference numbers):

The studies involving human participants were reviewed and approved by the medical ethics committee of Chongqing University Three Gorges Hospital (2022KY13).

## Data Availability

The data is available at.figshare: Fang, Chengmei (2025). Baseline.xlsx. figshare. Dataset. https://doi.org/10.6084/m9.figshare.28090235.v2.

## Clinical Trial Registration

The following information was supplied regarding Clinical Trial registration:

MR-50-24-013402.

## Supplemental Information

Supplemental information for this article can be found online at http://dx.doi.org/10.7717/peerj.19779#supplemental-information.

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
