# Peer review of "Serum β-klotho is a potential biomarker for diagnosing alcoholic liver disease and differentiating from nonalcoholic fatty liver disease"

_PeerJ, doi:10.7717/peerj.19779_

## Round 0.1 · original submission · Major Revisions

The reviewers have provided detailed and specific feedback.

·

Basic reporting

Dear Authors,

Thank you for submitting your manuscript titled "Serum β-klotho is a potential biomarker for diagnosing alcoholic liver disease and differentiating from nonalcoholic fatty liver disease" for consideration. I have carefully reviewed your work, and while I find the core premise of using sKLB as a novel biomarker for ALD diagnosis intriguing, several aspects of the manuscript require substantial revision before it can meet publication standards.

Your investigation addresses an important clinical challenge in differentiating ALD from NAFLD, and the potential of sKLB as a biomarker represents a potentially valuable contribution to the field. The preliminary data suggesting superior diagnostic performance compared to traditional markers like AST/ALT ratio and GGT is particularly interesting. However, the current presentation of this work does not fully capitalize on its potential significance.

The most pressing concerns relate to the methodological rigor and clarity of presentation. The statistical framework needs strengthening, particularly regarding sample size justification and the propensity score matching methodology. The discrepancy in reported sample sizes between different sections of the manuscript (207 versus 204) needs to be resolved. Additionally, the mechanistic underpinnings of sKLB elevation in ALD require more thorough exploration and discussion.

The manuscript would benefit significantly from a more robust presentation of the laboratory methods, particularly regarding sample handling and ELISA validation. The figures, while informative, need substantial revision to meet publication standards, especially in terms of statistical annotation and technical quality.

Experimental design

Introduction section:
Structural and Flow Issues
The introduction (lines 58-84) has several structural weaknesses that need to be addressed:
"With the effective prevention and control of infectious liver diseases such as viral hepatitis B(Liu et al., 2019) and C(Te & Jensen, 2010) as well as the change of lifestyles, the morbidity of alcoholic liver disease (ALD) and non-alcoholic liver disease (NAFLD) are growing" (lines 59-62) – This opening sentence attempts to cover too many concepts at once. The transition from infectious diseases to ALD/NAFLD needs more explanation. Consider breaking this into multiple sentences and explaining the epidemiological shift more clearly.

"studies have revealed the key role of KLB in regulating hepatic lipid and bile acid (BA) metabolism(Somm et al., 2018), but its role in the pathogenesis of ALD is unclear" (lines 77-79) - This statement highlights a knowledge gap but fails to explain why understanding this role is important. The authors should elaborate on how KLB's known functions in lipid metabolism might relate to ALD pathogenesis.
"In vivo KLB is produced by gene mutation or proteolytic cleavage in two forms: secreted KLB (also known as soluble KLB or serum KLB, sKLB, only extracellular fragments)(Lee et al., 2018) and membrane-bound KLB (mKLB, with both intracellular and extracellular fragments)(Kuzina et al., 2019), while sKLB role in alcoholic liver disease has not been reported" (lines 79-83) - This passage introduces technical details about KLB forms but doesn't clearly establish why studying sKLB in ALD is important. The authors should explain the theoretical basis for investigating sKLB specifically.
"GGT and AST/ALT ratios are often used to diagnose ALD and distinguish NFALD markers, the specificity is often insufficient" (lines 67-69) - This statement needs quantitative support from the literature regarding the specificity values. Additionally, the phrasing "distinguish NFALD markers" is unclear - do they mean distinguishing ALD from NAFLD?
"Therefore, this study was to analyze the role of sKLB in ALD and to explore the diagnostic value of sKLB in differentiating ALD from NAFLD" (lines 83-84) - This statement of purpose comes abruptly without sufficient build-up. The authors should better explain why sKLB might be a promising biomarker based on its biological functions and previous research.

Materials and Methods section
"A total of 70 ALD, 68 NAFLD and 66 healthy controls (HC) were enrolled from the Department of Hepatology, and Health Management Center" (lines 88-89) - The authors fail to:
• Justify their sample size through power calculations
• Describe their sampling strategy (consecutive, random, etc.)
• Explain how they accounted for potential selection bias between hospital and health center populations
• Detail the timeframe rationale for recruitment (May 2019 to August 2022)

"Diagnosing ALD relied on alcohol consumption exceeding 20 g/d for females and 40 g/d for males, drinking history > 5 years, or at least 80 g/d for 2 weeks" (lines 100-102) - This presents several methodological issues:
• No explanation of how alcohol consumption was verified
• Lack of standardized assessment tools for alcohol intake
• Absence of validation methods for self-reported drinking history
• No clear distinction between acute and chronic ALD cases
The healthy control group criteria require substantial clarification: "HCs were non-drinkers with no major health issues" (line 94) - This definition is problematically vague:
• "Non-drinkers" need precise quantification
• The term "major health issues" requires explicit definition
• The screening process for excluding health conditions needs a detailed description
• The matching criteria between controls and cases should be explained

The laboratory methodology section has critical omissions: "sKLB was detected by the ELISA kit" (lines 114-115) - The authors need to provide:
• Validation data for the ELISA assay
• Intra-- and inter-assay coefficients of variation
• Quality control measures
• Details about sample handling and storage conditions
• Information about batch effects and randomization of sample analysis

"Statistical analyses were performed using R4.4.0 and GraphPad Prism software" (line 125) - This section needs:
• Justification for the choice of statistical tests
• Methods for handling missing data
• Approaches for dealing with outliers
• Description of normality testing procedures
• Details about any data transformations performed

The methods lack crucial information for study reproducibility:
The authors should include:
• Detailed reagent specifications
• Equipment calibration procedures
• Standard operating procedures for sample collection
• Quality control measures
• Raw data availability statement

Validity of the findings

The discussion opens weakly and lacks a proper theoretical framework: "Currently, the diagnosis of ALD primarily depends on medical history(Kong et al., 2019)" (line 179). This opening fails to:
• Synthesize the key findings in the context of the research question
• Present a clear argument for why SKL represents a significant advance
• Connect the findings to existing theoretical understanding of ALD biomarkers
There is inadequate exploration of underlying mechanisms: "The underlying mechanism of sKLB elevation in patients with ALD is currently unknown" (line 239) - This crucial limitation is buried near the end of the discussion. The authors should:
• Address this limitation earlier
• Propose specific mechanistic hypotheses
• Discuss potential molecular pathways
• Connect their findings to known ALD pathophysiology
The clinical relevance discussion needs substantial strengthening: "sKLB emerged as a promising non-invasive serological marker for diagnosing ALD" (lines 192-193) - This claim requires:
• Comparison with existing clinical diagnostic algorithms
• Discussion of practical implementation challenges
• Cost-benefit analysis versus current methods
• Consideration of different clinical scenarios where sKLB testing might be valuable

The limitations section (lines 244-248) is superficial and needs expansion: "This study was conducted exclusively on an Asian population." The authors should:
• Discuss specific genetic or environmental factors that might affect generalizability
• Address potential selection bias in their sample
• Consider how their findings might differ in other populations
• Acknowledge the technical limitations of their methodology

The future research implications are inadequately developed: "increasing the female sample size is necessary to improve the study's rigor" (lines 246-248). This simplistic recommendation fails to:
• Outline a comprehensive research agenda
• Identify key knowledge gaps
• Propose specific follow-up studies
• Consider broader implications for ALD biomarker research
The discussion of statistical findings lacks sophistication: "The AUROCs of sKLB + AST/ALT, sKLB + GGT, and AST/ALT + GGT in diagnosing ALD were 0.924, 0.967, and 0.917, respectively" (lines 175-176) - The authors should:
• Discuss the clinical significance of these differences
• Address potential overfitting
• Consider validation requirements
• Examine effect sizes and confidence intervals

The discussion inadequately integrates existing literature: "Previous studies showed that KLB expression was reduced in MAFLD" (line 195) - The authors need to:
• Provide a more comprehensive literature review
• Critically evaluate conflicting findings
• Place their results in a broader context
• Identify areas of consensus and controversy

Additional comments

Figure 1: Expression Analysis Plots
Major Technical Issues: "Data are depicted as box-and-whisker plots." - The presentation has several problems:
• Statistical significance markers (*) are missing between the compared groups
• Y-axis scales vary between panels, making visual comparison difficult
• No indication of sample sizes (n) in the figure legend
• Missing a clear explanation of how outliers were determined and handled
• Box plot elements (median, quartiles) need a clearer definition in the legend
Visualization Concerns: Panel D shows AFLD, AC, and ALF groups, but:
• Error bars are inconsistent across groups
• Small sample sizes appear to affect the reliability of the boxplots
• No justification for the chosen data presentation format
• The color scheme could be improved for clarity and accessibility



Figure 2: ROC Curve Analysis
Methodological Issues: "The ROC curve demonstrates the diagnostic performance." This figure requires substantial revision:
• Confidence intervals should be visually represented on the curves
• Missing comparison with standard clinical cutoffs
• No indication of optimal cutoff points on the curves
• Axis labels lack units and clear descriptions
• The legend fails to explain the clinical significance of the AUROCs
Technical Presentation:
• Inconsistent formatting between panels
• Font sizes vary and may be too small for publication
• Missing gridlines to aid in visual interpretation
• No clear indication of how specificity and sensitivity were calculated.


Figure 3: Combined Biomarker Analysis
Statistical Representation Issues: "Receiver Operating Characteristic curve for sKLB + AST/ALT, sKLB + GGT and AST/ALT + GGT" - The analysis presentation needs improvement:
• No explanation of how the combined markers were integrated
• Missing statistical comparison between individual and combined markers
• Unclear methodology for determining combined marker performance
• Absence of validation of cohort results
• No discussion of potential overfitting

Reviewer 2 ·

Basic reporting

This study demonstrated that serum β-klotho (sKLB) is a potential biomarker for diagnosing alcoholic liver disease (ALD) and distinguishing it from nonalcoholic fatty liver disease (NAFLD). Notably, sKLB levels were significantly elevated in ALD and reduced in NAFLD, with improved diagnostic accuracy when combined with γ-GT.
Differentiating NAFLD from ALD is often a clinical challenge and an interesting aspect of this study; however, several concerns exist regarding this research.

Major points
1. KLB is believed to be associated with both NAFLD and ALD, but is the comparison between NAFLD and ALD in this study appropriate? How does KLB behave in other liver diseases, such as viral hepatitis, aside from healthy individuals? Please discuss this with reference to relevant literature if available.

2, sKLB is also reported to be associated with NAFLD, but why was it elevated only in the ALD group in this study? Please discuss the possible reasons for this finding.

3, sKLB is a potentially useful biomarker, but how does it compare to existing markers such as GGTP in terms of cost and rapidity? Please also discuss its applicability in clinical practice.

4. Although NAFLD and MAFLD are discussed in the Background, please note that they are not entirely synonymous; care should be taken in how these terms are used.

Minor points
1. Shouldn't the figures include explanations for the abbreviations used? Please check and ensure clarity.

2. There is a discrepancy in the reported sample size between the abstract and the main text (207 vs. 204). Please ensure consistency.

Experimental design

no comment

Validity of the findings

no comment

---

## Round 0.2 · accepted · Accept

Dear Dr. Wu,

Thank you for submitting the revised version of your manuscript. After a thorough evaluation of your revisions by Reviewers and me, I am pleased to inform you that all reviewer comments have been satisfactorily addressed. Accordingly, your manuscript is now accepted for publication in PeerJ.

Sincerely,
Stefano Menini

Reviewer 2 ·

Basic reporting

no comment

Experimental design

no comment

Validity of the findings

no comment